# A VGG-19 Model with Transfer Learning and Image Segmentation for Classification of Tomato Leaf Disease

**Thanh-Hai Nguyen \*** , **Thanh-Nghia Nguyen** and **Ba-Viet Ngo**

Faculty of Electrical-Electronic Engineering, Ho Chi Minh City University of Technology and Education, Ho Chi Minh City 700000, Vietnam

**\*** Correspondence: nthai@hcmute.edu.vn; Tel.: +84-906-738-806

**Abstract:** Tomato leaves can have different diseases which can affect harvest performance. Therefore, accurate classification for the early detection of disease for treatment is very important. This article proposes one classification model, in which 16,010 tomato leaf images obtained from the Plant Village database are segmented before being used to train a deep convolutional neural network (DCNN). This means that this classification model will reduce training time compared with that of the model without segmenting the images. In particular, we applied a VGG-19 model with transfer learning for re-training in later layers. In addition, the parameters such as epoch and learning rate were chosen to be suitable for increasing classification performance. One highlight point is that the leaf images were segmented for extracting the original regions and removing the backgrounds to be black using a hue, saturation, and value (HSV) color space. The segmentation of the leaf images is to synchronize the black background of all leaf images. It is obvious that this segmentation saves time for training the DCNN and also increases the classification performance. This approach improves the model accuracy to 99.72% and decreases the training time of the 16,010 tomato leaf images. The results illustrate that the model is effective and can be developed for more complex image datasets.

**Keywords:** VGG-19 model; tomato leaf images; transfer learning; image segmentation; HSV color space





## 1. Introduction

One major problem in the agriculture field is that harvest performance could be damaged by many different plant diseases. Tomato (*Solanum lycopersicum* L.) is one of the most important and popular vegetable crops in the world [1]. Thus, the prevention of tomato plant diseases has attracted scientists, aiming to increase the harvest performance [2]. There are more than a dozen different diseases of tomato plants in practice, so it is essential to detect them as accurately and as early as possible to prevent and treat the disease [3,4]. There are different methods of detecting plant diseases for different treatment, including those that use artificial intelligence (AI) algorithms of SVM based on image futures or neural networks (NNs) [5–7]. It is obvious that AI techniques have been applied in many fields of agriculture to identify plant diseases such as apple, tomato, and rice; others have used deep learning networks, which will be applied in this research for classifying tomato leaf diseases [8–11]. In our research, one deep learning network will be employed for classifying tomato leaf diseases.

In recent years, deep learning (DL) has been applied for the classification of plant conditions in order to find diseases for early treatment [12–16]. Significantly, the CNN was employed to improve the identification performance from 91% and 98% of 13 leaf diseases [17]. Liu at el. proposed using the CNN with AlexNet for classifying four apple leaf types: mosaic, rust, brown spot, and Alternaria leaf spot [18]. With this CNN, the diseases on apple leaves were detected with high recognition performance, up to 97.62%. In another study, Lu et al. utilized a deep CNN model to detect rice diseases using 500 images of 10 different common conditions [19]. This model achieved an accuracy of 95.48%, which is much higher than using a traditional machine learning network.

Tomato leaf diseases have attracted many researchers and different algorithms have been proposed for recognizing and classifying tomato diseases [20–22]. The combination of three convolutional network families for recognizing tomato diseases was proposed [20]. In particular, these faster region-based convolutional neural network (Faster R-CNN), region-based fully convolutional network (R-FCN), and single shot multibox detector (SSD) were combined to create the meta-architectures with "deep feature extractors" and this increased the classification performance. One of the famous datasets is the Plant Village database, consisting of 16,010 images of 10 tomato leaf diseases, which is often used in many research articles [23]. In [24], the authors proposed a novel PCA–whale optimization-based deep neural network model for classifying tomato plant diseases. In particular, the PCA-whale optimization was applied to extract features of the leaf images before being fed into the deep learning network model for classification. This model represented not only an outstanding performance but also difficulties in some cases related to the lacking number of samples.

CNN models and a transfer learning method have been applied for classifying tomato leaf diseases [25–28] to create a pre-trained model for increased prediction. In particular, the authors proposed two models using transfer learning and feature extraction in classifying tomato plant diseases and the obtained result was an accuracy of 90% [29]. Moreover, Rangarajan et al. [30] developed a pre-trained deep-learning algorithm with two AlexNet and VGG-16 models, in which the transfer learning was employed for classifying six tomato crop diseases and one healthy class in image sets obtained from the Plant Village database. This research showed high accuracies of 97.29% using VGG-16 and 97.49% using AlexNet, respectively. In another study [31], a CNN with the transfer learning approach was employed to recognize nine leaf disease sets, in which the automatic extraction of features by directly processing the raw images was performed. This method of the CNN with the transfer learning achieved a high accuracy of 99.18%.

Detection of leaf diseases can be performed by extracting features of plant leaf images or feature extraction and DL-CNNs are combined [32]. Image-processing methods such as enhancing, filtering, segmentation, thresholding, or feature extracting were applied with the back propagation network for detecting or recognizing leaf diseases and the classification result was very good [33–39]. In agriculture, plant diseases can significantly reduce the quality and quantity of products. In tropical regions, apples are widely grown and also attacked by pathogens or fungi such as bacterial, algal, and nematodes [40]. In this study, the authors proposed image segmentation based on color differences to separate the apple leaf disease areas. In particular, color (RGB, HSV) histogram, and texture (LBP) features are applied to extract feature vectors with rich information. After the feature extraction of leaf disease areas, advanced machine learning algorithms (fine KNN, SVM, bagged tree, complex, and others) are applied for recognition. The classification accuracy using bagged tree is 99%, with apple leaf disease. This is obvious proof that the reasonable image processing for the apple leaf image set combined with machine learning is effective.

Segmentation plays an essential role in detecting plant disease through different leaf conditions. In particular, in the proposed segmentation method using hybrid sub-types [41], the whole color leaf image was firstly divided into a number of nearly homogenous super-pixels creating super-pixel clusters, which could be helpful to clusters for image segmentation to increase convergence speed. The pixel lesions were quickly and accurately segmented using the expectation-maximization (EM) algorithm from each super-pixel. In another work, Singh et al. applied image segmentation and soft computing techniques to automatically detect and classify plant leaf diseases such as banana, beans, jack-fruit, lemon, mango, potato, tomato, and sapota [42]. This work achieved an average accuracy of 97.6% and this illustrates the effective proposed method.

One possible solution for segmenting leaf diseases was developed by Zhang et al. [43]. In this research, internet of things (IoT) was employed and the combination of clusters of super-pixels, K-means, and pyramid of histograms of orientation gradients (PHOG) was also proposed. In particular, the color images were split into many small super-pixels

using the clustering method. Therefore, the K-mean algorithm was to segment the lesion images from each super-pixel. Finally, three color components of each segmented part were combined with its gray-scale image to create four PHOG descriptors as one vector. This method showed an accuracy of 85.64% for the apple leaf image sets, including Alternaria, mosaic, and rust diseases. The group of Storey used the mask R-CNN algorithm for segmenting rust diseases on apple leaves; then, fractional masks on a subset of the Plant Pathology Challenge 2020 database were applied to produce the classification accuracy of 80.5% [44].

This article consists of four sections. Section 2 presents the structure of the VGG-19 model with transfer learning combined with image segmentation for classifying ten tomato leaf diseases, consisting of nine tomato leaf disease types and one healthy type. These images are segmented to extract the disease leaf areas and the same black background using HSV color space. In Section 3, the description of parameters adjusted to improve the model performance and reduce the training time of the VGG-19 is given, and the experimental results using the proposed method are discussed. Finally, Section 4 presents the conclusions about this research.

## 2. Materials and Methods

### 2.1. Plant Village Tomato Leaf Image Datasets

The tomato leaf image dataset (Plant Village) has 16,010 images with ten types [45]. Table 1 shows 9 types of tomato bacterial spot, tomato Septoria leaf spot, Mosaic virus, leaf mold, target spot, early blight, yellow leaf curl virus, tomato late blight, two-spotted spider mites, and one healthy image type. All images have different sizes and need to be resized into 224 × 224 or 256 × 256 for the input size of VGG-19. The number of each image type is imbalanced; for example, the yellow leaf curl virus disease has 3208 images, and the Mosaic virus only has 373 images. Therefore, this can be one of the reasons for decreasing the classification accuracy when applying the DCNN structure.

**Table 1.** The total tomato leaf images used in this research.

| No. | Class of Tomato Leaf Images | Images |
|---|---|---|
| 1 | Tomato bacterial spot disease | 2127 |
| 2 | Tomato Septoria leaf spot disease | 1771 |
| 3 | Mosaic virus disease | 373 |
| 4 | Leaf mold disease | 952 |
| 5 | Target spot disease | 1404 |
| 6 | Early blight disease | 1000 |
| 7 | Yellow leaf curl virus disease | 3208 |
| 8 | Tomato late blight disease | 1908 |
| 9 | Two-spotted spider mites | 1676 |
| 10 | Healthy leaf | 1591 |
| Total of tomato leaf images | | 16,010 |

### 2.2. Proposed Model for Classifying Tomato Leaf Diseases

As shown in Figure 1, the proposed model consists of a tomato leaf dataset, image segmentation, VGG-19 with transfer learning, and an evaluation block. The resized images are used to extract the leaf regions and the same black background from the HSV color space. In addition, the image features are applied for pre-trained layers through the transfer learning method. Finally, the model is evaluated by a confusion matrix.

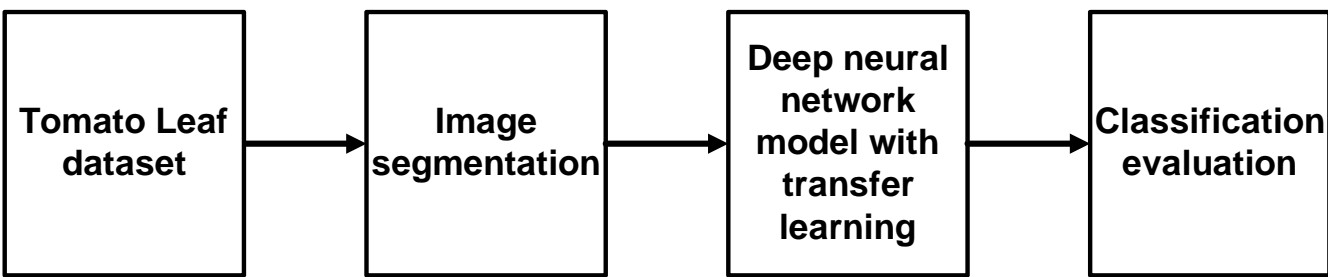

**Figure 1.** Proposed model for classifying tomato leaf disease and classification accuracy evaluation.

*2.3. Tomato Leaf Image Segmentation Using the HSV Color Space*

HSV is more meaningfully related to the psychological perception of color than RGB. Therefore, an HSV color space can separate color information from intensity or lighting. In addition, a histogram can be constructed for choosing a thresholding rule using only saturation (*S*) or hue (*H*). In practice, it is just a nice improvement that even by singling out only the hue, it still has a very meaningful representation of the base color that is much better when compared with RGB. As a result, there is more robust color thresholding over simpler parameters using the HSV color space.

For segmentation to extract leaf regions and the black backgrounds of all images, HSV of all leaf images can be uniformed perceptually, and all components of images can be quantized with the same precision. In particular, the HSV is a three-dimensional cartesian coordinate system, and its brightness values (*V*) can vary from 0 to 1. The hue (*H*) means that its color can range from 0 to 360 degrees, in which 0 or 360 degrees is red, 60 degrees of 60, 120, 180, 240, and 300 correspond to the colors of yellow, green, cyan, blue, and magenta, respectively. The saturation (*S*) defines the white color and is mixed with *H* to produce different colors represented by the percentage of the range from 0 to 1. The 1 means a pure color such as red, green, or blue. One image in the RGB color space can be transformed to the HSV using the following equations [46]:

$$H(x,y) = \begin{cases} 0^o, \ if \ \Delta = 0 \\ 60^o.\left( \frac{G'(x,y)-B'(x,y)}{\Delta} \bmod 6 \right), \ if \ T_{\max} = R'(x,y) \\ 60^o.\left( \frac{B'(x,y)-R'(x,y)}{\Delta} + 2 \right), \ if \ T_{\max} = G'(x,y) \\ 60^o.\left( \frac{R'(x,y)-G'(x,y)}{\Delta} + 4 \right), \ if \ T_{\max} = B'(x,y) \end{cases} \tag{1}$$

$$S(x,y) = \begin{cases} 0, \ if \ T_{\max} = 0 \\ \frac{\Delta}{T_{\max}}, \ if \ T_{\max} \neq 0 \end{cases} \tag{2}$$

$$V(x,y) = T_{\max} \tag{3}$$

in which, $R'(x,y) = \frac{R(x,y)}{255}$, $G'(x,y) = \frac{G(x,y)}{255}$, $B'(x,y) = \frac{B(x,y)}{255}$, $T_{\max} = \max(R'(x,y), G'(x,y), B'(x,y))$, $T_{\min} = \min(R'(x,y), G'(x,y), B'(x,y))$, and $\Delta = T_{\max} - T_{\min}$.

For the segmentation to extract leaf regions and to create the black backgrounds of all original RGB images, the algorithm is described as follows:

- **Step 1**: The RGB image is converted into the image with the HSV color space. The HSV components (three sub-images) can easily distinguish colors related to the type of color, the shade of color, the purity of color, or the brightness of color.
- **Step 2**: Histograms for all three HSV components are plotted to choose their lower and upper threshold values.
- **Step 3**: Masking is to segment and convert the HSV images to binary images based on the histogram HSV thresholds before extracting the leaf region. Therefore, we can fill holes in one binary image to create a leaf mask using the morphological operation.

- **Step 4**: The white mask can map to the RGB image for collecting the original leaf region and the black background.

### 2.4. VGG-19 Model

The DL networks can be applied for image classification in many fields based on large datasets with around 60 million parameters and 650,000 neurons [47]. In practice, the network architecture can have five convolutional layers and three fully connected layers with different roles. There are two first convolution layers (standard layer and max-pooling layer), the 3rd and 4th convolution layers (directly connected), the last convolution layer (max-pooling layer), and the output layer (softmax layer). In addition, some networks have particular architectures for unique applications. For instance, GoogleNet is a network with about 7 million parameters, 9 inception modules, 4 convolutional layers, 4 max-pooling layers, 3 average pooling layers, 5 fully connected layers, and 3 softmax layers [48]. All convolutional and dropout layers use the ReLU (activation function) with a parameter reduction ratio of 70% applied to all fully connected layers. Additionally, ResNet is similar to VGG-19 and has been adapted many times to produce ResNet-18, ResNet-34, ResNet-50, ResNet-101, and ResNet-152 [49]. Herein, we apply VGG-19 to train the tomato leaf image dataset.

Basically, VGG has an architecture of a CNN network, and VGG-19 is one of the VGG-based architectures [50]. The VGG-19 is a deep-learning neural network with 19 connection layers, including 16 convolution layers and 3 fully connected layers. The convolution layers will extract features of the input images, and the fully connected layers will classify the leaf images for those features. In addition, the max-pooling layers will reduce the features and avoid overfitting, as described in Figure 2.

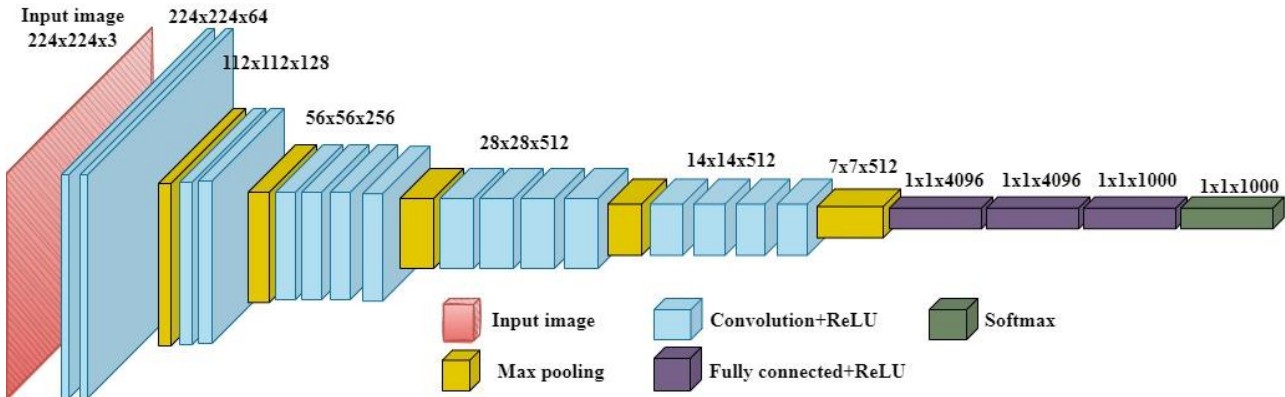

**Figure 2.** Representation of the VGG-19 architecture used in this research.

The output of each convolutional layer is represented by the following expression:

$$C_j^l = \varphi \left( \sum_{i=1}^{M^{l-1}} C_i^{l-1} \times k_{ij}^l + b_j^{l-1} \right) \tag{4}$$

in which $\times$ is the convolutional function which describes the connection between the weights of the *i*th and *j*th features in the $(l-1)$th, *l*th layers, $b_j$ is the bias value, and $\varphi$ is the activation function.

### 2.5. Transfer Learning

Transfer learning is often used in DL networks from the trained object to re-train the other objects. There are four types of transfer learning: case-based, features-based, parameter-based, and relationship-based. Obviously, choosing the trained parameters for the best classification system is a big challenge. Here, a suitable network architecture needs to be addressed along with the network parameters, and these values need to be

estimated for the new input data. Then, the new network needs to be fine-tuned to improve performance. In this paper, the parameter-based transfer learning method was applied for classifying tomato leaf diseases. In particular, the VGG-19 network will freeze the convolution layers and re-train the fully connected layers to enhance the classification. Furthermore, we also adjust the batch size, epoch, and learning rate to choose the best network.

### 2.6. Evaluation of Classification System

Using a confusion matrix (Table 2), we can evaluate the performance when classifying the 9 disease types and 1 healthy type of tomato leaf. Some parameters, such as accuracy (*ACC*), sensitivity (*SEN*), specificity (*SPE*), positive predictive value (*PPV*), and F1 score (*F1S*), are also considered in the VGG-19 model.

**Table 2.** Representation of the confusion matrix with many layers [51].

| | | **Predicted Classes** | | | | | | |
|---|---|---|---|---|---|---|---|---|
| | | 1 | ... | *i* | ... | *C* | | |
| | 1 | $n_{11}^T$ | ... | $n_{1i}^F$ | ... | $n_{1C}^F$ | $N_1$ | |
| | $\vdots$ | $\vdots$ | $\ddots$ | $\vdots$ | | $\vdots$ | $\vdots$ | |
| **True classes** | *i* | $n_{i1}^F$ | ... | $n_{ii}^T$ | ... | $n_{iC}^F$ | $N_i$ | |
| | $\vdots$ | $\vdots$ | $\ddots$ | $\vdots$ | | $\vdots$ | $\vdots$ | |
| | *C* | $n_{C1}^F$ | ... | $n_{Ci}^F$ | ... | $n_{Ci}^T$ | $N_C$ | |
| | | $P_1$ | ... | $P_i$ | ... | $P_C$ | $N_T$ | |

In the confusion matrix with the output at the *i*th layer, the value $n_{ij}^T$ is the number of correctly recognized images, and $n_{ij}^F$ describes the number of images at the *i*th layer but recognized to the *j*th layer, $(i \neq j)$. Furthermore, $P_i$ represents the total number of images at the *i*th layer after recognition, $N_i$ describes the total number of images at the *i*th layer in the original label, and $N_T$ is the total number of images in the testing dataset. Therefore, the $N_i$, $P_i$, and $N_T$ values are calculated using the following equations:

$$N_i = n_{ii}^T + \sum_{j \neq i} n_{ij}^F, \tag{5}$$

$$P_i = n_{ii}^T + \sum_{j \neq i} n_{ji}^F, \tag{6}$$

$$N_T = \sum_{i=1}^{C} N_i = \sum_{i=1}^{C} P_i \tag{7}$$

From the values of $N_i$, $P_i$, and $N_T$, the *ACC*, *SEN*, *SPE*, *PPV*, and *F1S* values are determined using the following formulas:

$$ACC = \frac{1}{N_T} \sum_{i=1}^{C} n_{ii}^T, \tag{8}$$

$$SEN = \frac{1}{C} \sum_{i=1}^{C} \frac{n_{ii}^T}{N_i}, \tag{9}$$

$$SPE = \frac{1}{C} \sum_{i=1}^{C} \frac{\sum\limits_{i \neq j} n_{ij}}{\sum\limits_{1}^{C} N_i - N_i} \tag{10}$$

$$PPV = \frac{1}{C}\sum_{i=1}^{C}\frac{n_{ii}^{T}}{P_i}. \tag{11}$$

$$F1S = 2 \times \frac{REC \times PRE}{REC + PRE} \tag{12}$$

The model can be evaluated through Equations (8)–(12), in which the classifier is highly effective when the *ACC*, *SEN*, *SPE*, *PPV*, and *F1S* are large.

## 3. Results

### 3.1. Image Segmentation Result

In this paper, the initial tomato leaf images were resized to the size of 224 × 224 as shown in Figure 3. Therefore, the segmentation method of the tomato leaf images with the same size was proposed for extracting the leaf regions and the same black backgrounds before being used for training the VGG-19. In particular, the HSV color space and histogram thresholds were applied for creating binary images before segmentation to produce the leaf regions and the same background.

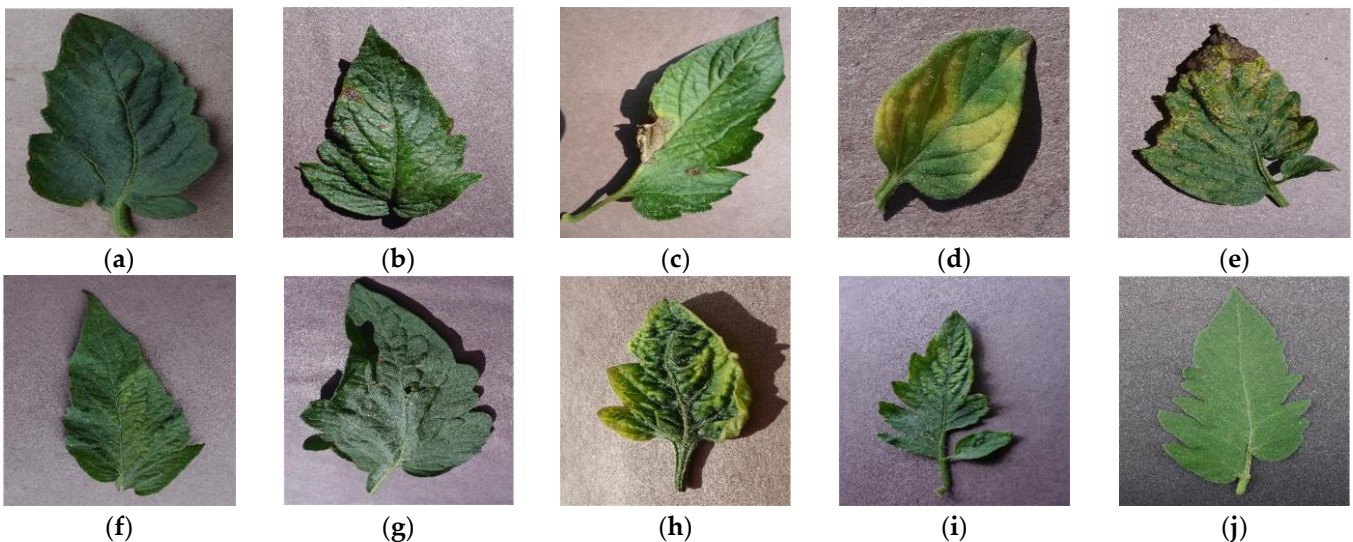

**Figure 3.** Ten samples of tomato leaf disease and healthy images: (**a**) Bacterial spot disease; (**b**) Early blight disease; (**c**) Late blight disease; (**d**) Leaf mold disease; (**e**) Septoria leaf spot disease; (**f**) Two-spotted spider mites; (**g**) Target spot disease; (**h**) Yellow leaf curl virus disease; (**i**) Mosaic virus disease; (**j**) Healthy.

Figure 4 shows the RGB image and the image in the HSV color space, in which the HSV image has three images with the H, S, and V components. The images with the HSV components were transformed to produce the corresponding histograms, called sub-images. Before plotting these histograms, we normalized the values of the components in the range from 0 to 1. The threshold values for H, S, and V were calculated based on the gray levels of the histograms as shown in Figure 5. From the histograms, we observed that the lower and upper threshold values for H are 0.18 and 0.5, due to the green colors of tomato leaves, and similarly, the values are 0.1 and 1.0 for S, and 0.0 and 0.8 for V.

The morphological reconstruction of one image was performed based on the mask image convoluted to the RGB input image (Figure 6a) to produce the binary image (Figure 6b). In addition, to produce the image with the leaf region and the black background as shown in Figure 6c, the binary image was multiplied with the RGB image.

Figure 7 shows the images after segmentation with the original leaf regions and the same black backgrounds, in which the leaves can retain the disease areas. Therefore, the dataset can speed up the training time, reduce many computations inside the neural network, and increase the accuracy.

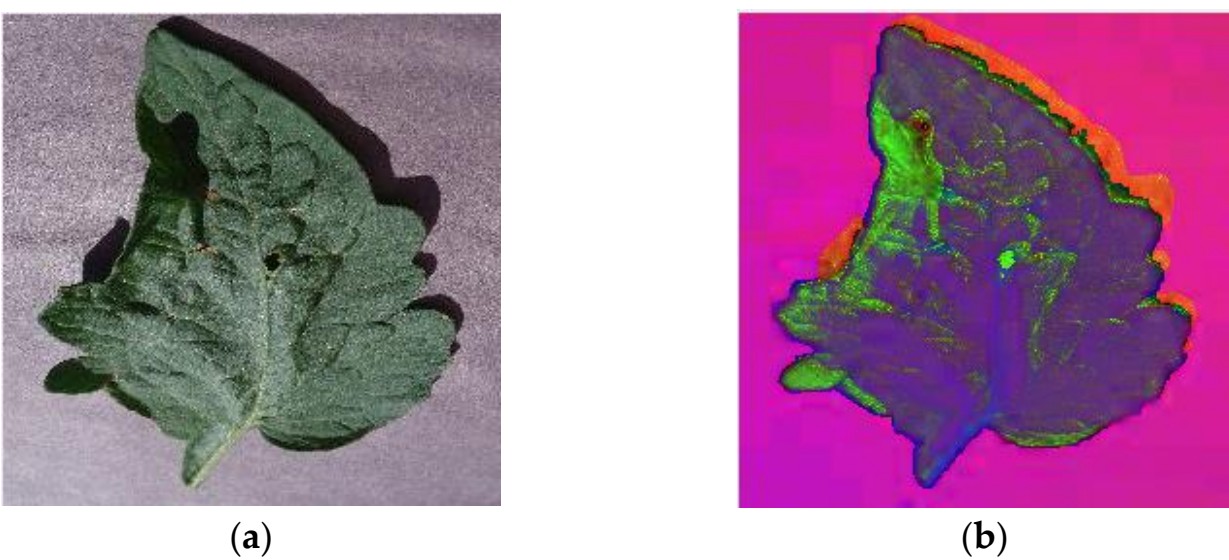

**Figure 4.** Representation of the processed image: (**a**) the RGB image; (**b**) the HSV image converted from the RGB image.

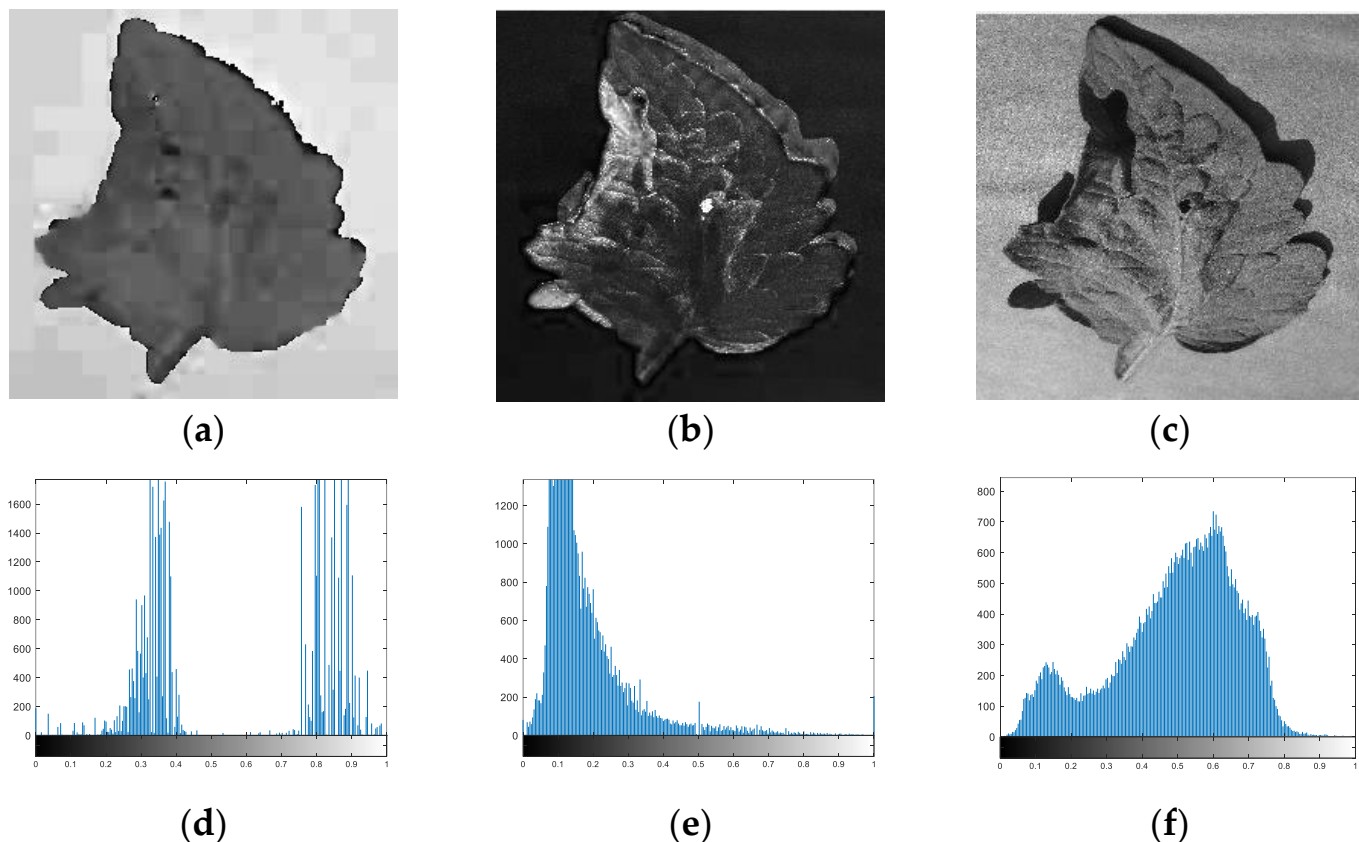

**Figure 5.** Input image in the HSV color space and the histograms of its components: (**a**) Hue component image; (**b**) Saturation component image; (**c**) Value component image; (**d**) Histogram of Hue component image; (**e**) Histogram of Saturation component image; (**f**) Histogram of Value component image.

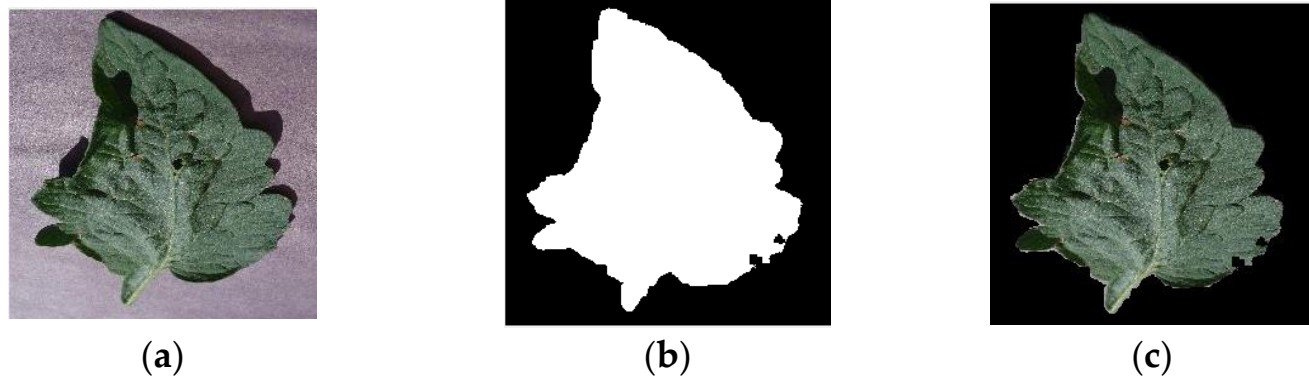

**Figure 6.** The results of the segmented image: (**a**) the RGB input image; (**b**) the binary image processed using one mask; (**c**) the image with the leaf region and the black background.

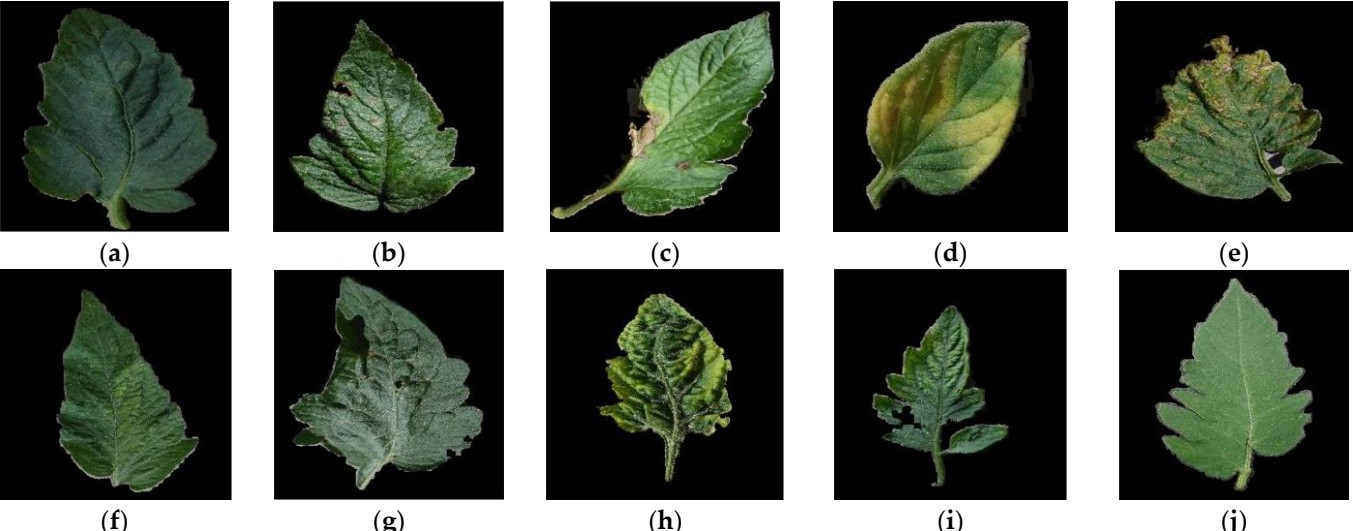

**Figure 7.** Ten segmented leaf images including nine leaf disease images and one healthy image: (**a**) Bacterial spot disease; (**b**) Early blight disease; (**c**) Late blight disease; (**d**) Leaf mold disease; (**e**) Septoria leaf spot disease; (**f**) Two-spotted spider mites; (**g**) Target spot disease; (**h**) Yellow leaf curl virus dis-ease; (**i**) Mosaic virus disease; (**j**) Healthy leaf.

Figure 8 describes five field tomato leaf images with the different diseases which have the different backgrounds (Figure 8a–e) and the segmented images to produce the leaf regions and the black backgrounds (Figure 8f–j). With the segmented field images using the HSV color space, the obtained results are similar to those of the images from the Plant Village database. In this research, we performed the segmentation of images based on the histogram thresholding values of the leaf regions, while the different backgrounds of the images are not relative to the segmentation results.

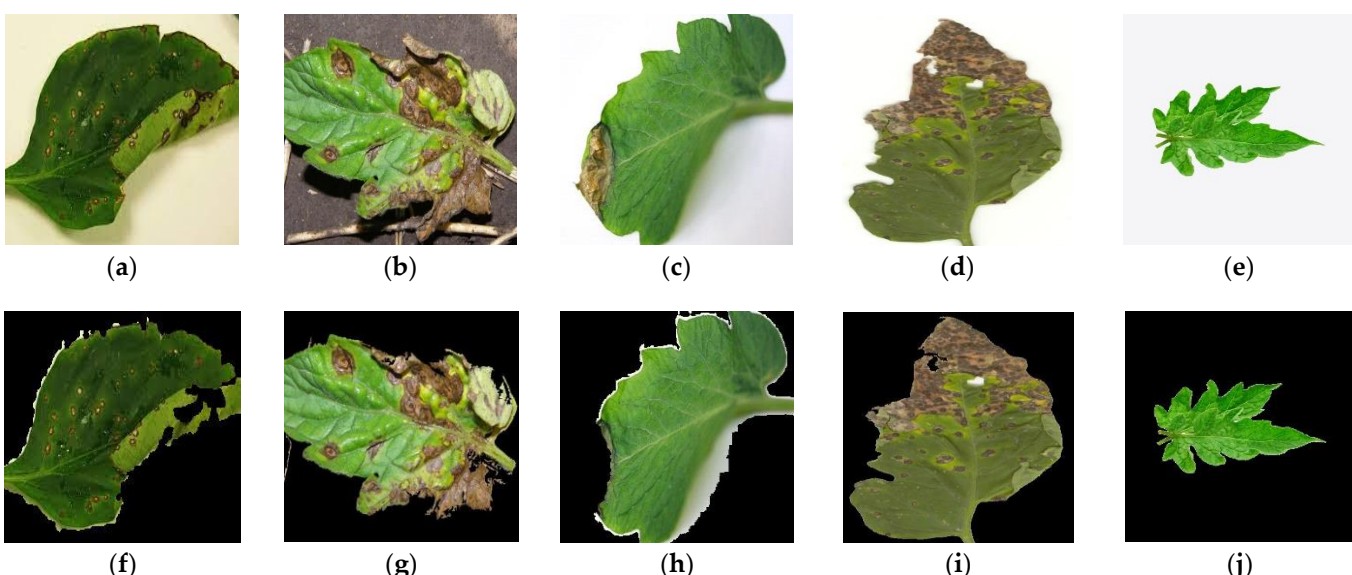

**Figure 8.** Five segmented field tomato leaf images: (**a**) The field images of bacterial spot disease; (**b**) Early blight disease; (**c**) Late blight disease; (**d**) Septoria leaf spot disease; (**e**) Healthy leaf; the segmented field images of (**f**) Bacterial spot disease; (**g**) Early blight disease; (**h**) Late blight disease; (**i**) Septoria leaf spot disease; (**j**) Healthy leaf.

To evaluate the performance of the HSV technique, we calculated a metric or structural similarity index (SSIM) between the HSV-segmented images and the manually segmented images [52]. The SSIM range was installed to be from 0 to 1. The maximum value of 1 indicates that the structurally segmented image is similar to the initial one, and inversely, the minimum of 0 is no structure, as described in Table 3.

$$SSIM(x, y) = \frac{(2\mu_x\mu_y + C_1)(2\sigma_{xy} + C_2)}{(\mu_x^2 + \mu_y^2 + C_1)(\sigma_x^2 + \sigma_y^2 + C_2)} \tag{13}$$

where $x$ is the original image and $y$ denotes the segmented image; $\mu_x$ and $\mu_y$ denote the mean intensity for the reference image $x$ and image $y$, respectively. In addition, $\sigma_x^2 = 1/N-1 \sum_{i=1}^{N}(x_i - 1/N \sum_{i=1}^{N} x_i)^2$ and $\sigma_y^2 = 1/N-1 \sum_{i=1}^{N}(y_i - 1/N \sum_{i=1}^{N} y_i)^2$ describe the standard deviation of the original and segmented images, and $\sigma_{xy} = 1/N-1 \sum_{i=1}^{N}(x_i - 1/N \sum_{i=1}^{N} x_i)(y_i - 1/N \sum_{i=1}^{N} y_i)$ refers to the covariance between the original and segmented images. $C_1 = (K_1L)^2$, and $C_2 = (K_2L)^2$ are constants, in which $K_1$ and $K_2$ should be less than 1 and $L$ is the number of gray levels of a grayscale image or an image in three channels, such as the RGB image.

**Table 3.** Description of the SSIM values between 10 segmented leaf images and the original leaf ones.

| No. | Image Name | SSIM |
|---|---|---|
| 1 | Bacterial spot disease | 0.914 |
| 2 | Early blight disease | 0.905 |
| 3 | Late blight disease | 0.929 |
| 4 | Leaf mold disease | 0.908 |
| 5 | Septoria leaf spot disease | 0.912 |
| 6 | Two-spotted spider mites | 0.946 |
| 7 | Target spot disease | 0.924 |
| 8 | Yellow leaf curl virus disease | 0.917 |
| 9 | Mosaic virus disease | 0.922 |
| 10 | Healthy leaf | 0.924 |

### 3.2. Classification Performance Using the VGG-19 with Transfer Learning

The VGG-19 is a deep and wide structure in which the number of computational parameters is well-optimized. In particular, the parameters were configured for training the network, including epochs (300), hidden layer active function (Tansig), output active function (Softmax), initial learning rate (0.00001), and batch size (60).

In this research, the division of 16,010 tomato leaf images (100%) in the Plant Village dataset is performed as follows: 80% of the image set for training and validation, and 20% of the image set for testing. In addition, the images for training and validation are divided into the training (80%) and the validation (20%).

Figure 9 demonstrates the confusion matrix of the model; where 1 is bacterial spot disease, 2 is early blight disease, 3 is late blight disease, 4 is leaf mold disease, 5 is Septoria leaf spot disease, 6 is two-spotted spider mites, 7 is target spot disease, 8 is yellow leaf curl virus disease, 9 is Mosaic virus disease, and 10 is the healthy leaf, respectively. We observed that the SEN value of most leaf diseases is high, at over 99%, except the SEN value of the mosaic virus type, at 97.4%. This result can be because the mosaic virus dataset is only 373 images, compared with the other datasets from 1000 to 3000 images. To improve, we can apply the up-sampling technique to increase the number of datasets or the balanced class weight method to enhance the accuracy of the mosaic type. Furthermore, the PPV value of most diseases is perfective at 100%. So, the model has an average accuracy of 99.71%. This value demonstrates that the proposed model is very effective for classifying the tomato leaf diseases.

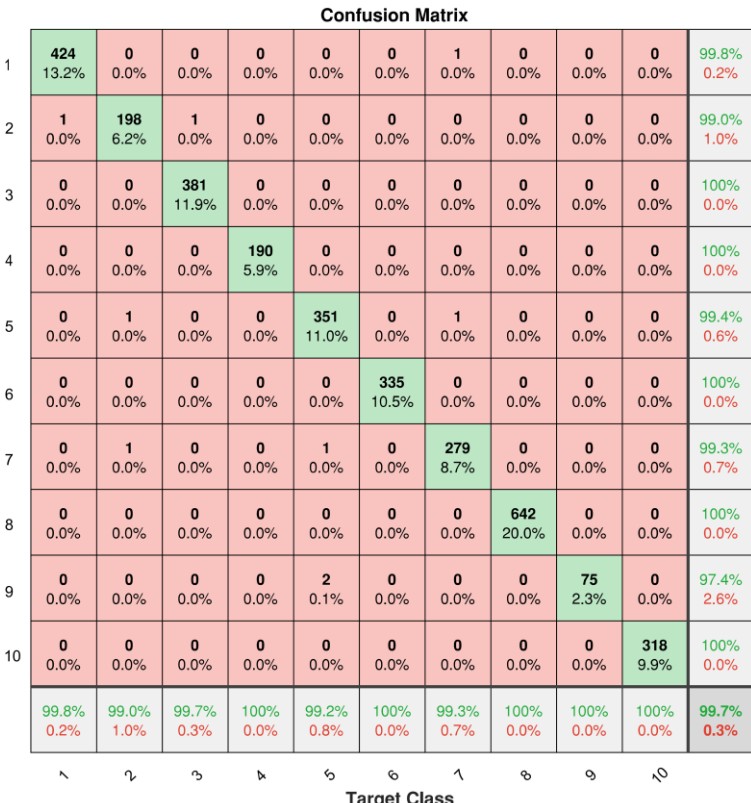

**Figure 9.** Classification evaluation results of nine leaf diseases and one healthy leaf: (**1**) Bacterial spot disease, (**2**) Early blight disease, (**3**) Late blight disease, (**4**) Leaf mold disease, (**5**) Septoria leaf spot disease, (**6**) Two-spotted spider mites, (**7**) Target spot disease, (**8**) Yellow leaf curl virus disease, (**9**) Mosaic virus disease, (**10**) Healthy leaf.

### 3.3. Effect of Learning Rate Parameter on Classification Performance

We fine-tuned the initial learning rate (LR) value to investigate the suitable training parameters and then evaluated the output performance. The result of each value is shown in Table 4. In particular, the obtained classification accuracy is 99.72% for LR at 0.00001, 99.34% for LR at 0.0001, and 99.02% for LR at 0.001, respectively. The model performance decreases when increasing the LR value because the system does not reach the best convergence point. Therefore, the best LR is 0.00001 for the classifier of the tomato leaf diseases.

**Table 4.** Classification evaluation results with the confusion matrix parameters and three different learning rates (in unit %).

| Learning Rate | 0.00001 | 0.0001 | 0.001 |
|---|---|---|---|
| ACC | 99.72 | 99.34 | 99.02 |
| SEN | 99.69 | 99.36 | 98.48 |
| SPE | 99.90 | 99.77 | 99.66 |
| PPV | 99.49 | 99.23 | 98.75 |
| F1S | 99.59 | 99.29 | 99.00 |

### 3.4. Effect of Epoch Parameter on Classification Performance

Epochs describe the number of training times of the neural network until the training is stopped. The model will not match the training data (under-fitting) when the epoch is too small, and the model will be over-fitting when this value is too large. In both cases, the classification result is not good. However, we cannot calculate the suitable epoch and we must choose this value based on the model and dataset. Table 5 shows the chosen epoch values from 50 to 400. We observed that the accuracy is highest at 99.72% at 300 epochs. Furthermore, the accuracy is 98.50% at 50 epochs because the number of trained times is not enough (under-fitting). Inversely, the accuracy is 99.66% at 400 epochs because the number of trained times is too much (over-fitting). In addition, Figure 10 shows that the epoch is chosen to be 300, the loss function value gradually approaches the minimum value. From those results, choosing 300 epochs represents the best value for achieving optimal performance.

**Table 5.** Classification evaluation results with confusion matrix parameters and five different epochs (in unit %).

| Epochs | 50 | 100 | 200 | 300 | 400 |
|---|---|---|---|---|---|
| ACC | 98.50 | 99.13 | 99.53 | 99.72 | 99.66 |
| SEN | 98.61 | 99.18 | 99.54 | 99.69 | 99.59 |
| SPE | 99.32 | 99.66 | 99.83 | 99.90 | 99.89 |
| PPV | 98.16 | 98.75 | 99.38 | 99.49 | 99.43 |
| F1S | 98.39 | 98.97 | 99.46 | 99.59 | 99.51 |

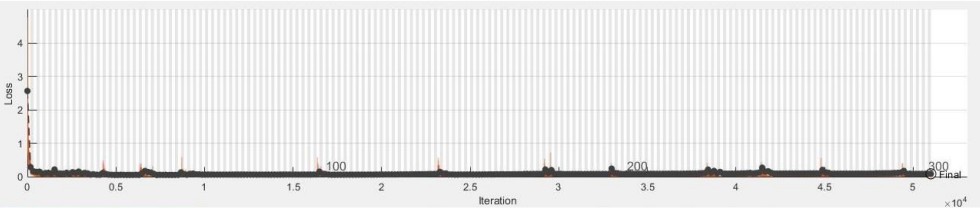

**Figure 10.** The representation of the loss function values during training the network using 300 epochs.

### 3.5. Comparison of Performance Results with Difference Models

Different models were applied to investigate tomato leaf disease classification, such as AlexNet, GoolgeNet, and ResNet50. Most network architectures gave relatively good

results, as shown in Figure 11. In this experiment, we used the tomato leaf image dataset divided into the sub-datasets for training and testing, as shown in Section 3.2. With the same parameters, the AlexNet has an accuracy of 99.16%, and the ResNet has an accuracy of 99.19%. The GoogleNet shows an accuracy slightly higher than the two above models, at 99.38%. Meanwhile, our model achieved the highest accuracy at 99.72% with VGG-19.

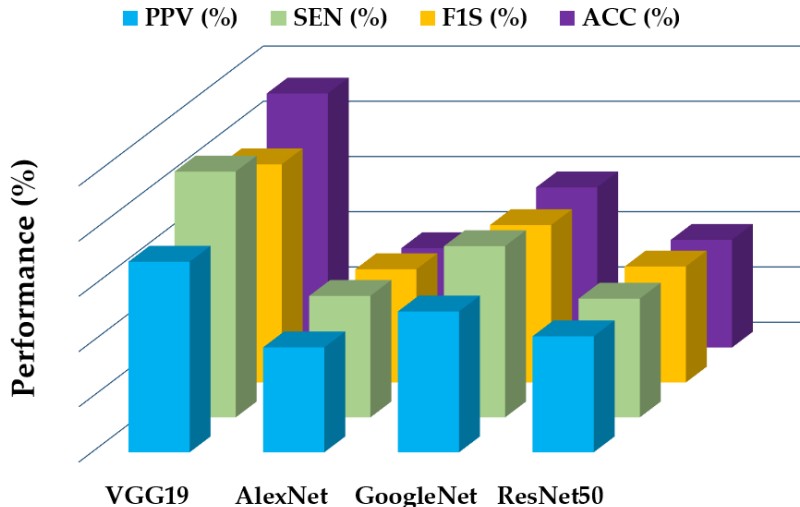

**Figure 11.** Classification performance comparison of tomato leaf diseases using four different deep-learning network architectures.

### 3.6. Comparison of Time for Training and Testing between Segmented and Non-Segmented Images

In this paper, we also investigate the leaf disease classification performance and time between non-segmented and segmented images. In particular, Table 6 presents the results of the leaf disease classification performance as well as time for training and testing between the two sets of non-segmented and segmented leaf disease images using the HSV color space method. The performance of leaf disease classification when using the segmented method is higher than that of using the non-segmented one. In particular, the classification accuracy of leaf diseases in the two cases are 99.72% and 99.63%, respectively. It is obvious that time for training and testing of the leaf disease classifier in the case of the segmented images is faster than that in the non-segmented ones; specifically, it takes the network using the segmented images $2.75 \times 10^5$ s (training) and 29.30 s (testing), while times of the network using the non-segmented images are $2.98 \times 10^5$ (training) and 35.27 (testing). The reason for this achievement of better performance and time is that the network using the segmented images only focuses on the feature region of the leaf image and the same background of the leaf images is black. Meanwhile, for the original image, the network is trained for both the leaf image region and background.

**Table 6.** Comparison of time for training and testing between non-segmented images and segmented images.

| Preprocessing | ACC (%) | SEN (%) | SPE (%) | PPV (%) | F1S (%) | Training Time (second) | Testing Time (second) |
|---|---|---|---|---|---|---|---|
| Segmented images | 99.72 | 99.69 | 99.90 | 99.49 | 99.59 | $2.75 \times 10^5$ | 29.30 |
| Non-segmented images | 99.63 | 99.58 | 99.82 | 99.37 | 99.47 | $2.98 \times 10^5$ | 35.27 |

## 4. Discussion

The experimental results of this article were compared with previous works, in which the Plant Village dataset of tomato leaf diseases was used in all deep-learning network models. Indicators related to pre-processing, transfer learning, model, classes of images, and other parameters are described in Table 7.

**Table 7.** Tomato leaf disease comparison between previous methods and results.

| Work | Model | Transfer Learning | Classes | Preprocessing | ACC |
|---|---|---|---|---|---|
| Maeda-Gutiérrez et al. [53] | GoogleNet | Yes | 10 | No | 99.39 % |
| Brahimi et al. [31] | GoogleNet | Yes | 10 | No | 99.35 % |
| Agarwal et al. [28] | CNN model | No | 10 | No | 98.40 % |
| Gadekallu et al. [24] | MLP | No | 10 | PCA-WOA | 94.00 % |
| Trivedi et al. [54] | CNN model | No | 10 | Transformed into grey images | 98.49% |
| Our proposed model | VGG-19 | Yes | 10 | HSV color space for image segmentation | 99.72 % |

Without transfer learning, the features can be extracted from convolutional layers in the network or extracted by traditional feature extraction methods. Gadekallu et al. [24] proposed the principal component analysis–whale optimization algorithm (PCA-WOA) method for extracting features and then used it in a multilayer perceptron (MLP) network. This network has one input layer, two hidden layers, and one output layer for classifying ten different tomato leaf diseases. The accuracy of the obtained classifier was 94.00%. Agarwal et al. [28] and Trivedi et al. [54] developed convolutional layers in the CNN architecture for feature extraction and fully connected layers for disease classification. These studies applied pre-processing and transforming images to achieve model accuracies of 98.40% and 98.49%, respectively. In addition, the RGB images were pre-processed to produce the same size and grey images. This research only focused on applying the CNN, in which features of colors, textures, and edges were extracted for training. With transfer learning, Maeda-Gutiérrez et al. [53] and Brahimi et al. [31] proposed the GoogleNet architectures for classifying ten disease types. Their model accuracies were 99.39% and 99.35%, respectively.

Compared with the above studies, we optimized the tomato leaf dataset by segmenting to extract the leaf region and the black background. It is obvious that the segmentation of the leaf images reduced the training time and increased the classification performance. In addition, compared with previous research [24,28,31,53,54] without transfer learning, our VGG-19 model applied the transfer learning method with a learning rate of 0.00001, and epochs of 300; the classification results are outstanding, particularly with PPV at 99.49%, SEN at 99.69%, F1S at 99.59%, and ACC at 99.72%, respectively.

Finally, in Table 7 we find that the previous studies basically have the same proposed methods such as using transfer learning, ten classes, and a little bit of difference of the network model, GoogleNet [31,53] and VGG-19 (our model). The large difference is that our research applied HSV color space for image segmentation and the two studies [31,53] did not apply preprocessing for the image sets. The result is that the classification accuracy in our proposed method is a little bit higher, particularly the ACC value is 99.72% compared with 99.35% and 99.39%, respectively.

**5. Conclusions**

This paper proposed a classification model using the tomato leaf images segmented for training the VGG-19 with the transfer learning method. This classification model used nine disease types and one healthy type obtained from the Plant Village database which were segmented to extract the original leaf regions and the black backgrounds using the HSV. In addition, the learning rate and epochs were selected in the VGG-19 with the transfer learning to have the best classification network. In particular, the classification accuracy is 99.72%, higher than that of previously proposed works which had the highest ACC at 99.35% and lowest ACC at 94%. Moreover, the training time of the VGG-19 with the segmented images (2.75 s) and (29.30 s) is faster compared with that of the VGG-19 without the image segmentation (2.98 s and 35.27 s). The experimental results using the segmented leaf images and the best choice of the network parameters demonstrated the effectiveness of the proposed model. Moreover, the model with the leaf image segmentation and the VGG-19 architecture with transfer learning can be developed to be applied for classifying other leaf image datasets.

**Author Contributions:** Conceptualization, methodology, simulation, and writing original draft, B.-V.N. and T.-N.N.; methodology, supervision, validation and writing—review and editing, T.-H.N. All authors have read and agreed to the published version of the manuscript.

**Funding:** This work is supported by Ho Chi Minh City University of Technology and Education (HCMUTE) under Grant No. T2021-51TĐ.

**Data Availability Statement:** The data presented in this study are openly available at https://www.kaggle.com/datasets/emmarex/plantdisease.

**Conflicts of Interest:** The authors declare no conflict of interest.

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
