# Peer review of "A VGG-19 Model with Transfer Learning and Image Segmentation for Classification of Tomato Leaf Disease"

_agriengineering, doi:10.3390/agriengineering4040056_

Round 1
Reviewer 1 Report
The ms agriengineering-1862813 with the title of A VGG-19 Model with Transfer Learning and Image Segmentation for Classification of Tomato Leaf Disease investigates an interesting topic, but the ms has to be significantly improved before it can go for further process:
Please add the Latin name of tomato in first mention in introduction section, this is important not.
Please cite this ref in line 27-28: Mukhtar, T.; Rehman, S.u.; Smith, D.; Sultan, T.; Seleiman, M.F.; Alsadon, A.A.; Amna; Ali, S.; Chaudhary, H.J.; Solieman, T.H.I.; Ibrahim, A.A.; Saad, M.A.O. Mitigation of Heat Stress in Solanum lycopersicum L. by ACC-deaminase and Exopolysaccharide Producing Bacillus cereus: Effects on Biochemical Profiling. Sustainability 2020, 12, 2159. https://doi.org/10.3390/su12062159
L37-43 please add suitable citations for this text.
L55-59 please add suitable citations for this text.
L95-108 please add suitable citations for this text.
The introduction should be shorter than the current version, and authors have to cite the uncited text. Also, authors should remove common text and make it deeper.
Please add the hypothesis of this work at the end of the introduction section.
L142 leave space: PlantVillage
Figures 4 and 6: titles should be expanded and detailed.
Figure 10: very bad presentation, the authors should remove the values from above columns and from the bottom of Figure.
The discussion is very short and superficial written, the authors should make it deeper and stronger than the current version. In addition, they should add the most important values in the conclusion.
Regards, Reviewer
-
Reviewer 2 Report
Comments on “A VGG-19 Model with Transfer Learning and Image Segmentation for Classification of Tomato Leaf Disease”
Plant disease diagnostics is a critical step for effective and efficient management of crop losses with minimal or no damage to ecological balance. Traditionally disease was diagnosed with naked eye which calls for expertise. Even then, errors occur while quantifying the disease intensity due to inherent human error. Right through the years of computer revolution, efforts are underway to utilize computer-aided diagnostics. With the advent of deep learning systems, neural networks and similar novel tools, tools are being developed for accurate diagnostics.
This paper also provides enhancing the capabilities of such tools by introducing the transfer learning and image segmentation approaches. VGG-19 is a powerful tool and could provide viable solutions. However, my observations are:
1. Already published literature is available on transfer learning (Moyazzoma et al., 2021) and image segmentation (Vijai Singh et al., 2016; and use of VGG-19 (Hassan et al., 2021). Also, there are other papers. Unfortunately, the authors have not cited some of these papers also in the review.
2. The methodology and results are not in sync. Many details of methodology are missing whereas corresponding results are presented.
3. While the current paper used Plant Image database, the above papers have gone a step ahead to use field images and validate. This paper also should have collected some field images also as study material.
4. Specificity is one of the important parameters for correct diagnosis and it is not included as one of the parameters in the evaluation of classification system.
5. The authors have mentioned that 300 epoch is optimized value. If the validation loss also was accounted for, it would have justified the overfitting aspect.
6. Can we always apply segmentation under field conditions for the diagnostics. If so, what is the alternative.
7. Why HSV colour scheme was specifically used against other methods is not highlighted.
8. SSIM was calculated between HSV segmented images and the manually segmented images. But, in methodology, this component is missing.
9. Table 4 can be put in a running text.
10. Similarly, fig 8 also can be in running text.
11. Please specify the data used for comparison of different models (3.5).
12. Please mention the benefits due to transfer learning in terms of improvement in skill during training and converged skill of the trained model.
13. The authors have not mentioned which transfer learning technique was used by them though they mentioned 4 types at 2.5.
14. The discussion appears to be very weak and it can be elaborated already published literature not cited in this paper.
15. Authors may also review the language, especially the tense for results.
16. A few unwanted hyphenations distract reading and may be avoided.
17. Original Ms with a few suggestions for incorporation is also attached herewith please.
Hence, the paper should address all these issues before it is accepted for publication.
Round 2
Reviewer 1 Report
The ms has been improved, but the authors should add the Latin name of tomato in L27, directly after the English name (Solanum lycopersicum L.)
Reviewer 2 Report
The authors have complied with the points raised. In the edited version where new text has been added, English corrections need to be taken up.
